# Frailty in Kingdom of Saudi Arabia—Prevalence and Management, Where Are We? [note 1]

**DOI:** 10.3390/healthcare11121715

**Published:** 2023-06-12

**Authors:** Asim Muhammed Alshanberi

**Affiliations:** 1Department of Community Medicine and Pilgrims Health Care, Umm Alqura University, Makkah 24382, Saudi Arabia; amshanberi@uqu.edu.sa or asim.alshanberi@bmc.edu.sa; Tel.: +966-555-533-389; 2Medicine Program, Batterjee Medical College, Jeddah 21442, Saudi Arabia

**Keywords:** elderly population, frailty, geriatrics, Kingdom of Saudi Arabia, management, prevalence

## Abstract

A recent report from the United Nations state that the percentage of elderly individuals in the Kingdom of Saudi Arabia (KSA) will witness a sharp increase in the next three decades (5.6% in 2017 to 23% by 2050). This situation will lead to an increased prevalence of comorbidities and hence, will require close monitoring and constant care of such individuals who are prone to suffer from complications such as arthritis, cardiovascular disorders, diabetes, neurological disorders, etc. Frailty is one such age-related phenomenon which enhances the risk of falling, functional restrictions and greater vulnerability to adverse consequences, which tend to lead to institutionalization. Such factors highlight the importance of the urgent awareness for circumventing the progression of frailty toward a compromised health status. This concise report is an attempt to sum up the relevant research articles published with regard to frailty and concomitant diseases in the last 5 years. It also sums up the research on frailty in the KSA elderly population, till date. This article reflects the opinions of an author on tackling such issues through a well-directed mechanism involving interdisciplinary transitional care and geriatric co-management.

## 1. Frailty—Definition and Models Used for Its Prediction

Frailty is theoretically defined as an age-related syndrome characterized by the decline in physiological status of an individual, which leads to negative health outcomes such as fatigue and weakness, and a reduced tolerance to medical interventions. Currently, there is no gold standard for the definition of frailty [1]. The tools which are used widely for measuring frailty includes the *Fried’s frailty model* (or the frailty phenotype, Fried’s frailty criteria) and the *accumulation of deficits model*, which exploits the Frailty Index (FI) for characterizing frailty as a state [2]. Fried’s frailty phenotype is operationally interpreted by any three out of five phenotypic conventionalities, delineating compromised energetics, i.e., weight loss, exhaustion, reduced physical activity, slowness and weakness (Figure 1). A pre-frail stage is confirmed when such 1–2 criteria are fulfilled, which lead to the frailty stage.

## 2. Association of Frailty with Concomitant Diseases

Frailty is related to demographic factors, such as being female [3], increasing age [4], and the presence of adverse health events such as polypharmacy [5,6,7], decreased cognitive status [8], sarcopenia [9,10], falls [11,12], institutionalization, hospitalization [13] and death [14]. A detailed study was carried out earlier to determine the demographic and health factors related to the frailty syndrome in older adults [15]. Figure 2 shows that concomitant diseases such as cognitive impairment, pain, diabetes, poor sleep, hearing dysfunction, depression and a history of fall increases the risk of frailty among the geriatric community.

Frailty and diabetes mellitus (DM) are associated with increased disability and mortality in old age. Hence, prognosis and relevant treatment design varies greatly to diabetic elderly population observing frailty. The management of such individuals with type 2 diabetes mellitus (T2DM) is complicated by reduced life expectancy and adverse side effects from treatment and comorbidities. Such individuals are more inclined to hypoglycemic situations and related repercussions [16]. It has been recently reported that more than half of the population suffering from DM fall in the age group ≥ 65 years [17] and frailty prevalence is 3–5 times greater among such a population [18,19].

Moreover, a detailed article concluded that loss of hearing enhances the chances of frailty in old age [20,21]. In order to determine this, a prospective cohort study was achieved by Tian and co-workers [22] among a community sample of men aged 70 years and above. A total of 3285 participants who were free of frailty at the beginning of the study were followed for up to 17 years. It was observed that 71.5% of men developed frailty during follow up. Several longitudinal and cross-sectional studies have reported the association between frailty and cognitive disorders such as dementia and mild cognition [23,24]. Frailty leads to dementia and mild cognitive disorder in cognitively unimpaired populations [25]. Researchers have earlier explored the effect of psychological distress on associations between sleep quality and frailty in an elderly population suffering from osteoporosis and chronic diseases [26,27]. It was observed that frailty significantly impacted psychological distress and hence, the regular monitoring of psychological disorders and sleep problems in earlier stages would prove useful in the prevention of frailty among elderly individuals.

Falls represent a complex multifactorial phenomenon that leads to morbidity and disability in the elderly. The relationship between frailty and falls is complex, with one leading to the other in a vicious cycle. In this context, a study was aimed to determine the occurrence of sustained frailty remission and how remission is associated with falls risk [28]. Similarly, Ge and co-workers have examined whether physical frailty onset before, after, or in concert with cognitive impairment is differentially associated with fall incidence in community-dwelling older adults through a longitudinal observational study [29]. The research conducted to establish the link between frailty and cognitive impairment (CI) concluded that older adults experiencing “CI-frailty co-occurrence” had the greatest risk of repeated falls compared with those with “CI first” and “frailty first”, and this phenomenon is exhibited more pronouncedly in older women [30]. Pain in the elderly population is mainly related to neurodegenerative and musculoskeletal conditions, peripheral vascular diseases, arthritis and osteoarthritis that lead to poor life quality, social isolation, impaired physical activity and dependence to carry out daily activities. The consequences of untreated or inadequately treated pain are significant and may also lead to decreased immune functioning and cognition [31]. These pain-related consequences have a close relationship with and are similar to those found in older adults with frailty [32,33]. Moreover, frailty was found to be associated with depressive symptoms [34]. Mastaleru and co-workers (2022) investigated the link between depression and frailty among the geriatric population in a retrospective study. It was pointed out that frailty and depression impacted the quality of life negatively [35]. Hence, poor acknowledgment of these conditions and the underdiagnosis of such problems represent an important public health issue. Therefore, in order to improve the survival and successful aging of the elderly population, an active primary prevention protocol monitoring the factors contributing to frailty in early stages would be of paramount importance.

A thorough correlation of vitamin D with frailty in the elderly population was revealed in a recently appeared systematic review [36]. People with lower serum levels of 25-hydroxyvitamin D [25(OH)D] are more susceptible to becoming frail, although the impact of the regular intake of vitamin D on frailty is unknown. Cai and colleagues (2022) have recently monitored a randomized controlled trial through STURDY (study to understand fall reduction and vitamin D in you) among 688 adults above 70 years of age having low levels of 25(OH)D and an elevated risk of fall. The following characteristics were used to define the frailty phenotype: weakness, fatigue, slowness, poor activity and unintentional weight loss. It was concluded that a greater dosage of vitamin D was unable to decrease frailty [37].

Antidepressant usage is prevalent among the aged population; however, limited research witnessed its association with frailty [38]. An eight-week randomized placebo-controlled trial was performed by this group of researchers to monitor the relationship of antidepressant medication with frailty in the elderly population suffering from depression. It was determined that antidepressant medication is ineffective for treating major depressive disorder in elderly frail patients. Additionally, even when an antidepressant response is attained, this response only marginally lessens their vulnerability. These findings imply the need for novel therapeutic approaches to treat comorbid frailty and depression, as well as the inclusion of frailty measurements in regular psychiatric assessments of depressed older persons. Another study observed the consequences of antidepressants on the components of frailty. The findings revealed that the group that took antidepressants had a 62.7% prevalence of frailty. A connection between the user group and fatigue, slow walking speed and unintentional weight loss was discovered among the frailty components. Tricyclic antidepressants and potentially inappropriate medications (PIMs) that may not be suitable for elderly individuals were found to increase the risk of frailty in such a population. These findings show the urgency for a clinical assessment of the benefits and risks of prescribing antidepressants to the geriatric population. They also reported that once treatment is started, the evaluation and monitoring of geriatric characteristics are necessary to ensure the elderly’s safety and quality of life [39]. Moreover, C-reactive protein, hysterectomy and low free testosterone levels have all been linked to a greater risk of frailty among postmenopausal females [40,41]. Kim and Lee [42] observed the linkage between menopausal hormone therapy (MHT) and frailty through a cross-sectional study. This study underlined the detrimental connections between MHT and frailty in terms of treatment length between 2 and 5 years, starting before the age of 60, and the initiation of treatment within 3–6 years postmenopause. Hence, MHT may prove effective when given to postmenopausal females along with a proper treatment window to promote the healthy aging of such females.

## 3. Research on Frailty in KSA until Present Day

### 3.1. Frailty Prevalence

The global prevalence of frailty was noted as 40 cases per 1000 person-years, and also points out that the incidence rate was more in females as compared to males [43]. Moreover, as per a recent UN report, the percentage of the elderly Saudi population will rise to 23% by 2050 [44]. The government has paid great attention to provide care for elderly individuals [45] and researchers are playing their part in analyzing the prevalence of frailty in their population (Table 1). Alqahtani and co-workers have recently reported a 40% frailty prevalence among the Saudi population above 60 years [46]. A similar group has successfully executed a cross-sectional study in a tertiary care hospital in community-dwelling older adults for validating the Arabic version of the FRAIL scale [47]. The original FRAIL scale was expressed into Arabic (FRAIL-AR) and psychometric characteristics were assessed for each item on the FRAIL-AR scale over two visits within a seven-day interval. Fried frailty index (FFI) was taken as a reference measure to assess criterion-related validity. FRAIL-AR scale exhibited excellent internal consistency and reliability within this time period. The optimal cutoff point of 3 for frailty on the FRAIL-AR scale yielded a specificity of 67% and sensitivity of 72%. Moreover, frailty prevalence according to FFI and FRAIL-AR was noted as 28% and 37%, respectively. Additionally, Alqahtani et al. (2021) also evaluated the prevalence of frailty via Fried’s frailty phenotype by investigating the link between sociodemographic features and clinical factors, with frailty among 486 community-dwelling elderly individuals. Odds ratio (OR) and confidence intervals (IC) were used to analyze the findings via a multinomial logistic regression model. They observed the prevalence of frailty and pre-frailty as 21% and 47%, respectively [48].

### 3.2. Frailty and Sarcopenia

Another study witnessed the association of frailty with sarcopenic elderly patients [49]. This descriptive cross-sectional study involves the participation of 498 patients from a public tertiary hospital. The questionnaire includes questions related to activities of daily living and strength, assistance in walking, rise from a chair, stair climbing and falls (SARC-F), demographic data and the Edmonton frail scale. The prevalence of patients with severely frail, moderate frail and mild frail was observed as 4, 12 and 22%, respectively. It was revealed that frailty varies significantly with age, educational level, activities of daily living, the presence of comorbidity, marital status, patients’ needs of home care and sarcopenia. Sarcopenia exhibited a linear association with frailty in terms of several sociodemographic components of elderly KSA patients. Similarly, a protocol was developed (data generation and analysis is currently underway) by Alghannam and co-workers to reveal the prevalence of sarcopenia among such individuals in relation to their lifestyle behaviors [50]. The study aims to: (1) define Saudi reference values, which will prove fundamentals in diagnosing, treating and preventing sarcopenia; (2) determine cutoff points for muscle strength and physical performance; and (3) understand the modifiable lifestyle factors in the younger population. A two-step questionnaire was devised by Almufarrih and co-workers (2021) for evaluating frailty among 228 elderly patients attending the healthcare center. The former part of the questionnaire evaluated the medications of patient, demographics and comorbidities, whereas the latter comprises of Mini-Cog test and FRAIL scale. The detailed analysis revealed that males represented 52% of the studied sample and 42% of these were of 60–65 years of age, and frailty prevalence was observed as 25% [51]. Another study witnessed the prevalence of frailty among patients exhibiting cardiac stress via the Fried clinical frailty scale [52]. The frailty was found to be associated with a greater prevalence of cardiovascular risk factors such as hypertension and diabetes, while prevalence was recorded as 40%.

### 3.3. Frailty and Falls

In another cross-sectional survey targeting elderly Saudi citizens above 60 years, a one-year prevalence of falling was noted as 50% [53] and post-fall injuries were observed in 74% participants who experienced falls. Al-Ali et al. (2021) has recently performed a pilot study with 78 participants, which were divided into three categories; group-I, the control group, which included 31 healthy men (65–75 years), group-II, which included 25 patients with diabetes mellitus 2 (50–64 years) and group-III, which included 22 patients with diabetes mellitus 2 (65–80 years). FFI (Cardiovascular Health Study index) and FRAIL scale were used for evaluating diagnostic accuracy. It was observed that frail patients were more in groups II (44%) and III (55%) in contrast to group-I (10%), which were non-diabetic [54]. The prevalence of poor sleep quality and its association with the frailty status was analyzed through the Arabic version of Pittsburg sleep quality index (PSQI) and FFI [55]. The prevalence of frailty was observed as 37%. The prevalence of frailty in the elderly population undergoing PIMs was assessed for elderly inpatients in a public tertiary hospital [56]. Beers’ criterion was used for exploiting the data on a number of medications and PIMs use, whereas the FRAIL scale was exploited for analyzing the frailty status. The prevalence of frail and pre-frail patients was reported as 58% and 37%, respectively. A notable difference in frailty score was demonstrated by the PIM group as compared to the non-PIM group. Conclusively, PIM was associated with frail older adults with polypharmacy and multiple comorbidities. Alkeridy et al. [57] reported the linkage of frailty with poor health outcomes among home health care service (HHCS) users. This retrospective study concluded that frailty serves as a powerful predictor of mortality for a population receiving HHCS. Similarly, a cross-sectional study reported pre-frailty and frailty prevalence as 41% and 52%, respectively, among elderly Saudi patients [58].

## 4. Futuristic Management Plan—*Author’s Opinion*

Researchers have reported the association of frailty with the development of disabilities such as delirium, falls, mild cognitive impairment, incontinence and risk of hospitalization through well-defined screening tools. However, the focus should proceed to the designing of culture-friendly geriatric frailty screening tools in Arabic, which should be tested in a larger cohort. The recently developed Tilburg frailty indicator (TFI) by Gobbens and co-workers [59] will be used for assessing multidimensional frailty among community-dwelling older people. Despite being widely used, the TFI is not frequently utilized in Arabic-speaking nations, and there are not many Arabic-translated surveys that are effective at identifying and measuring frailty in the elderly population in Saudi Arabia. In order to predict the entire TFI and its physical, psychological and social domains for unfavorable outcomes such as disability, indications of healthcare consumption and falls, TFI will therefore be applied to a large portion of the population.

The management of frailty must be tailored to each patient’s goals of care and life expectancy. Physical exercises should be suggested to pre-frail patients, while palliative care and symptom control should serve as the core of management plan for patients exhibiting severe frailty. Health promotion, proper diet, social engagement and mild physical activity can also serve as an excellent vehicle to prevent frailty. Elucidating the etiology and natural history is therefore critical for identifying high risk subsets for the prevention and treatment of frailty. A comprehensive assessment tool may thus refine the quality of drug treatment in the elderly community, particularly who are associated to PIMs. Artificial intelligence and machine learning models can be used to predict frailty in a generalizable study cohort. Lastly and most importantly, research on this subject should involve common minds via collaborative mega-research projects for achieving the desired goal.

## 5. Conclusions

Frailty is a common condition in the elderly population, which can be easily identified by the frailty phenotype criteria. Since there is no universally employed screening tool for analyzing frailty, investigations are thus underway to determine the best method for identifying frailty. In the context of Arabic nations, the authors hold the opinion that Arabic versions of frailty scales can help in troubleshooting these issues. Frailty is associated with a number of adverse outcomes, many of which may be targets of treatment. Measures are required by healthcare providers and policymakers for intervention and the treatment of such individuals suffering from common disorders such as diabetes, dyslipidemia, heart failure, hypothyroidism, polypharmacy, cardiovascular, sleeping problems, etc. These basic steps can enhance the knowledge of clinicians in anticipating the robustness of such elderly frail patients.

## Figures and Tables

**Figure 1 healthcare-11-01715-f001:**
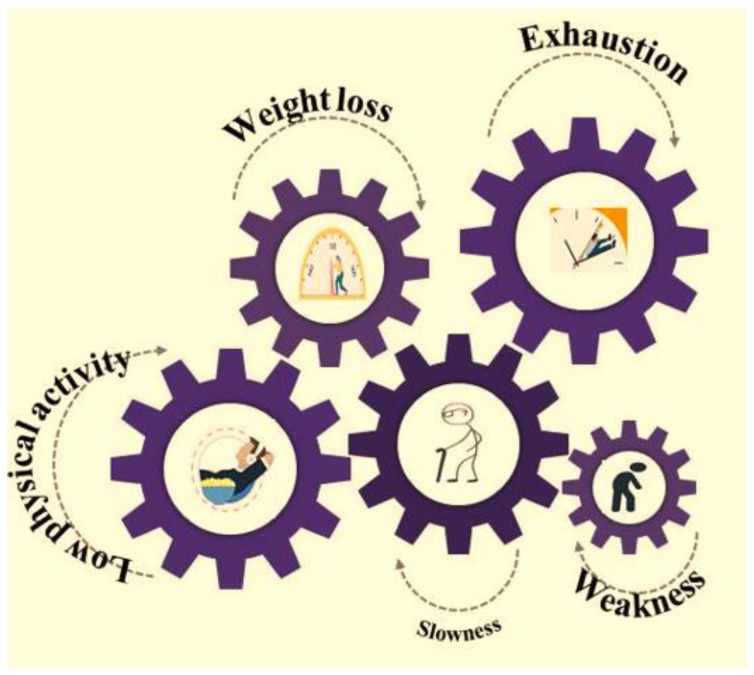
Components used to assess frailty according to the frailty phenotype.

**Figure 2 healthcare-11-01715-f002:**
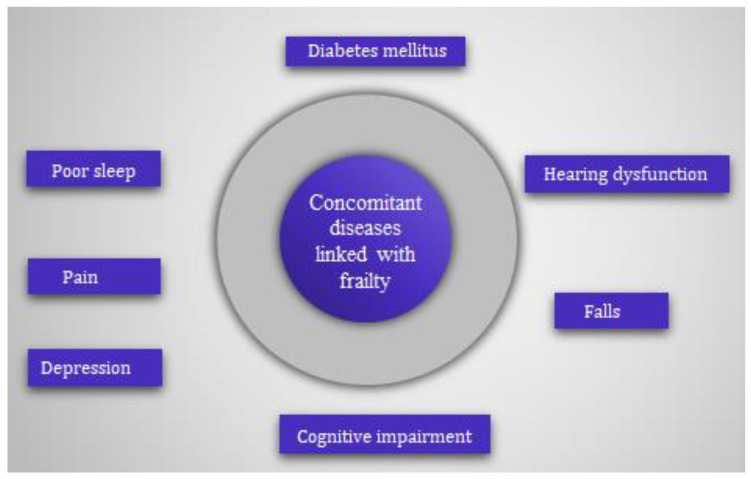
Concomitant diseased linked with frailty.

**Table 1 healthcare-11-01715-t001:** Estimation of frailty prevalence—scale used by KSA researchers.

S. No.	Type of Study	Scale	Setting	Sample Size	Reference
1	Cross-section	FFI	Outpatient	47	[47]
2	Cross-section	Fried frailty phenotype	Residence	486	[48]
3	Descriptive	Edmonton frail and SARC-F scale	Residence	498	[49]
4	Analytical	Global physical activity questionnaire, PSQI, Euro QOL five-dimensional questionnaire	Lifestyle and health research center at Princess Norah bint Abdulrehman University, Riyadh	1532	[50]
5	Observational	FRAIL scale and Mini-Cog test	Outpatient	228	[51]
6	Descriptive	Fried clinical frailty scale	Cardiac patients	876	[52]
7	Cross-sectional study by convenient sampling	Data collection form	Questionnaire	1182	[53]
8	Cross-section	Cardiovascular Health study index and FRAIL	Outpatient	78	[54]
9	Cross-section	PSQI and FFI	Community-based	270	[55]
10	Retrospective	Beers and FRAIL	Inpatient	358	[56]
11	Retrospective	Clinical frailty scale	King Saud University Medical Complex, Riyadh	555	[57]
12	Analytical	FRAIL	King Saud University Medical Complex, Riyadh	367	[58]

## Data Availability

Not applicable.

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
