# Peer review of "Frailty in Kingdom of Saudi Arabia—Prevalence and Management, Where Are We?†"

_healthcare, 2023, doi:10.3390/healthcare11121715_

Round 1
Reviewer 1 Report
Overall, the idea of the paper is very good and it can contribute to the evidence on frailty research in the Kingdom of Saudi Arabia.
Please see my comments as follows:
1. There any several major mistakes in describing the frailty definitions. For example, the Clinical Frailty Scale was developed by Professor Kenneth Rockwood and colleagues (in Canada) and to my best understanding, it is never called the Fried Clinical Frailty Scale. Professor Linda Fried (from the USA) is the one who developed the Fried’s frailty criteria (or frailty phenotype) which includes the 5 components of weight loss, exhaustion, reduced physical activity, slowness and weakness.
2. Page 1, paragraph 1: I suggest the author rewrite it like this:
Currently there is no gold standard for frailty definition. [1] The tools which are used widely for measuring frailty includes the Fried’s frailty model (or the frailty phenotype, Fried’s frailty criteria) and the Accumulation of Deficits model which exploits Frailty Index (FI) for characterizing frailty as a state [2]. The Fried’s frailty phenotype is operationally interpreted by any 3 out of 5 phenotypic conventionalities delineating compromised energetics i.e. weight loss, exhaustion, reduced physical activity, slowness and weakness (Fig 1). A pre-frail stage is confirmed when such 1-2 criteria are fulfilled, which leads to frailty stage.
3. Fig 1: should be renamed to “Components used to assess frailty according to the frailty phenotype”
4. The part “Association of frailty with concomitant diseases” is too lengthy and it aims to provide an overview on the evidence of the association between frailty and concomitant diseases in the world before the author move on to describe the frailty research in the Kingdom of Saudi Arabia. I suggest the author shorten it. Your article is to describe frailty research in the Kingdom of Saudi Arabia.
5. The author should describe how they came up with the studies (did the author conduct any systematic search on databases such as MEDLINE, EMBASE, or any local database in the Kingdom of Saudi Arabia)
6. In the “Research on frailty in KSA till now”, the author should provide sub-heading for when summarizing the studies related to frailty the Kingdom of Saudi Arabia.
Eg.
Frailty prevalence
Frailty and sarcopenia
Frailty and falls
Frailty and diabetes
7. Some other minor mistakes: Please explain abbreviation before use, and they should be consistent (eg. DM, T2DM, CI-cognitive impairment, PIM?).
I have no comments.
Author Response
The author appreciates the critical comments raised by the esteemed reviewers for the manuscript " Frailty in Kingdom of Saudi Arabia – prevalence and management, where are we?" (ID: healthcare-2391882) submitted to Healthcare journal. The changes have enhanced the quality of the manuscript. All the changes have been marked in red color.
Rev#1
- Agreed and re-written as suggested in point #2
- Done
- Done
- Please be informed that the first draft of the submitted article described only “Frailty research in the Kingdom of Saudi Arabia” but the matter was extremely less for publication (approximately 1100 words) because of only 13 publications on this topic from Saudi Arabia. However, as instructed by the esteemed academic editor, to increase the word count beyond 2500 words, to publish it as an Opinion, addition of this much material becomes a mandatory. Hence, relevant and recent work was cited for making it worth a publication.
- The author googled “frailty in Saudi Arabia” and found only 13 publications. Finding less research on this concerning subject makes the author curious of knowing the global pattern on frailty. In pursuit of this, most updated (last 5 years) literature survey was carried out to seek knowledge and information. Alternatively, the author is making a repository by using PubMed and Web of Science for relevant articles from January 2019 till December 2023 (last 5 years only). Pooled estimates are underway through random effect models and Mantel-Haenszel weighting. Homogeneity and risk of bias will be assessed before sending it to the journal for review process and publication.
- Sub-headings added as per your valuable suggestions.
Frailty and diabetes – Not given a sub-heading as it was connected to falls, and hence included in previous sub-heading.
- Done
Reviewer 2 Report
Comments to the Author
1. General comments
This paper is an overview of the frail and its current status and future in the Kingdom of Saudi Arabia. It was well written and easy to understand. To help the reader understand, please consider the following.
2. Specific comments
i) Several abbreviations are used in the paper, many of which are not spelled out. Please spell it all out for the first time.
ii) I felt that the word did not match the figure in the gear in Figure 1. Please confirm.
iii) I agree that frail is a very complex construct and the scale is not standardized. Several scales have been introduced, but I recommend a more comprehensive introduction to this point.
iv) Later in the paper, KSA's frail status is described. It is easy to understand, but I think we can understand the characteristics of KSA better if there is a slight comparison with other regions (Europe, North America, East Asia, etc.). Consider adding more information.
v) I agree with the part in the Authors Opinion about developing new scales that reflect culture. In the case of KSA, I would like to know exactly what scale it will be. I also think that too much specialization in a particular culture makes it difficult to compare with other countries, so how do you see that?
Author Response
Rev#2
- Specific comments
- i) Done
- ii) Edited, thanks for this in-depth comment.
iii) The author firly believe that the recently developed Tilburg Frailty Indicator (TFI), by Gobbens and coworkers [Gobbens RJ, Boersma P, Uchmanowicz I, Santiago LM. The Tilburg Frailty Indicator (TFI): New Evidence for Its Validity. Clin Interv Aging. 2020 Feb 21;15:265-274. doi: 10.2147/CIA.S243233. PMID: 32110005; PMCID: PMC7041595] may work and hence will be used for assessing multidimensional frailty among community-dwelling older people.
This point has been mentioned in the Futuristic management plan – Author’s opinion
- iv) It has been mentioned in answering the fifth comment of reviewer#1 that author is getting the information for this concerning subject in detail in the form of meta-analysis construction. So, it would be a separate manuscript.
- v) Done
Round 2
Reviewer 1 Report
I have nothing to add.
Author Response
Thank you so much Sir for your acceptance.
It has motivated me.